# Realizing thermoelectric cooling and power generation in N-type PbS$_{0.6}$Se$_{0.4}$ via lattice plainification and interstitial doping

Lei Wang[1], Yi Wen[1], Shulin Bai[1,2], Cheng Chang[1], Yichen Li[1], Shan Liu[1], Dongrui Liu[1], Siqi Wang[1], Zhe Zhao[1], Shaoping Zhan[1], Qian Cao[3], Xiang Gao [4], Hongyao Xie [1] ✉ & Li-Dong Zhao [1,2] ✉

Thermoelectrics have great potential for use in waste heat recovery to improve energy utilization. Moreover, serving as a solid-state heat pump, they have found practical application in cooling electronic products. Nevertheless, the scarcity of commercial Bi$_2$Te$_3$ raw materials has impeded the sustainable and widespread application of thermoelectric technology. In this study, we developed a low-cost and earth-abundant PbS compound with impressive thermoelectric performance. The optimized n-type PbS material achieved a record-high room temperature *ZT* of 0.64 in this system. Additionally, the first thermoelectric cooling device based on n-type PbS was fabricated, which exhibits a remarkable cooling temperature difference of ~36.9 K at room temperature. Meanwhile, the power generation efficiency of a single-leg device employing our n-type PbS material reaches ~8%, showing significant potential in harvesting waste heat into valuable electrical power. This study demonstrates the feasibility of sustainable n-type PbS as a viable alternative to commercial Bi$_2$Te$_3$, thereby extending the application of thermoelectrics.

Thermoelectric (TE) materials have garnered significant attention due to their capability of converting heat into electricity directly and reversibly[1–6], which has great potential to use for waste heat recovery to improve the energy utilization and sustainability. Additionally, TE cooling technology plays a crucial role in heat dissipation for electronic products, due to its high reliability and ease of miniaturization[7–10]. To assess the performance of TE materials, a dimensionless figure of merit $ZT = S^2\sigma T/\kappa_{tot}$ was used. It is defined by the Seebeck coefficient $S$, electrical conductivity $\sigma$, total thermal conductivity $\kappa_{tot}$ (including electrical thermal conductivity $\kappa_{ele}$, and lattice thermal conductivity $\kappa_{lat}$) and Kelvin temperature $T$. A good thermoelectrics need to possess a large $S$ and high $\sigma$, together with a low $\kappa$. However, these physical parameters $\sigma$, $S$ and $\kappa_{tot}$ are inherently interconnected through carrier concentration $n$, which makes the TE performance optimization be challenging[11–15].

As one of the earliest discovered thermoelectric semiconductors, Bi$_2$Te$_3$-alloy has dominated the commercial thermoelectric material for over half a century[3]. After years of continuous research, currently, the *ZT* of commercially applied Bi$_2$Te$_3$ materials is close to 1.0 at room temperature (Supplementary Fig. 1). However, in recent years, with in-depth research on this material system, it has been found that there is limited room for improvement in thermoelectric performance of Bi$_2$Te$_3$ materials[3,6]. Additionally, the small bandgap of Bi$_2$Te$_3$, approximately 0.15 eV, results in intrinsic excitations occurring around 50 °C, leading to a sharp decrease in thermoelectric performance with increasing temperature and limiting its application in thermoelectric power generation[16].

Moreover, as shown in Fig. 1a, tellurium is an extremely scarce element with a crustal abundance of only 0.001 ppm[17]. The global annual production of Te is currently only about 470 tons[18]. The rapid

[1]School of Materials Science and Engineering, Beihang University, Beijing 100191, China. [2]Tianmushan Laboratory, Hangzhou 311115, China. [3]Huabei Cooling Device Co. LTD., Hebei 065400, China. [4]Center for High Pressure Science and Technology Advanced Research (HPSTAR), Beijing 100094, China. ✉e-mail: xiehongyao@buaa.edu.cn; zhaolidong@buaa.edu.cn

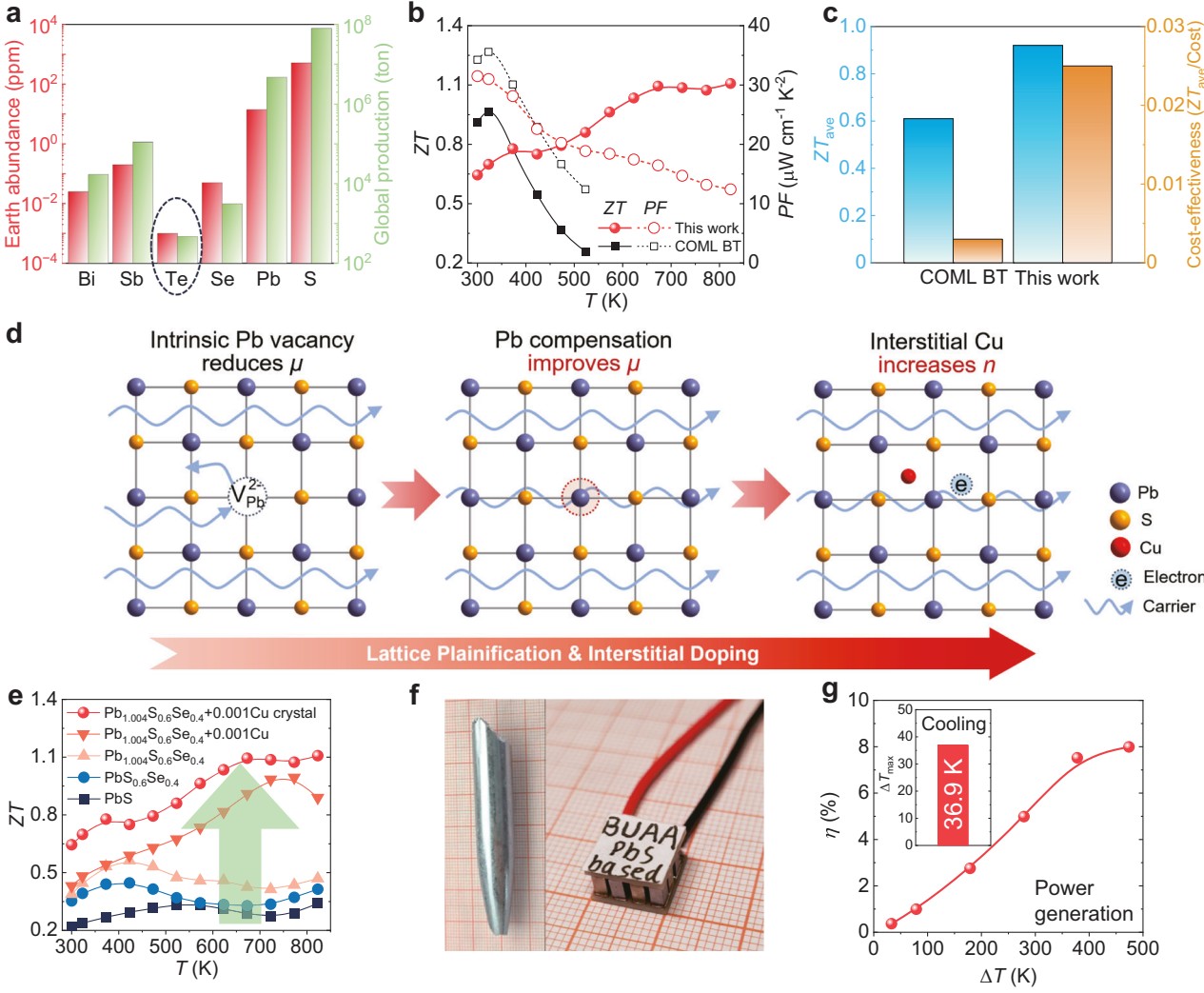

**Fig. 1 | High performance and earth-abundant PbS-based material demonstrates the TE cooling and power generation potential. a** The Earth abundance and global production of the constituent elements in commercial $Bi_2Te_3$ and PbS. **b**, **c** The comparison of $ZT$ value, power factor, $ZT_{ave}$ and cost-effectiveness ($ZT_{ave}$/cost) between commercial $Bi_2Te_3$ (COML BT) and our developed sample. **d** Schematic illustration depicting the progresses of lattice plainification and interstitial Cu doping for PbS. **e** The temperature-dependent $ZT$ showing the gradually increase in TE performance. **f** The pictures of high quality PbS crystal and the TE module based on n-type PbS and commercial p-type $Bi_2Te_3$ developed in this work. **g** The cooling effect $\Delta T_{max}$ of the PbS-based TE module and the power generation efficiency $\eta$ of the $Pb_{1.004}S_{0.6}Se_{0.4}$ + 0.001Cu single-leg device.

development of thermoelectric technology has driven a surge in demand for tellurium raw materials, which, in turn, limits the further widespread application of thermoelectric technology. Therefore, it is imperative to develop earth-abundant and high-performance thermoelectric materials that can replace $Bi_2Te_3$.

Lead chalcogenide, including PbTe, PbSe and PbS, is a group of traditional medium-temperature TE materials with impressive performance[19–35]. Among them, PbS has drawn wide attention because of its decent TE performance and earth-abundant composition (depicted in Fig. 1a)[17,36]. Because of the strong chemical bonding and light constitution of sulfur, pristine PbS possesses an intrinsic high $\kappa_{lat}$ and low $n$, which hinders its development in thermoelectrics[21]. To address these issues, numerous strategies have been developed to optimize the transport properties of PbS, such as forming solid solution[37], nano-structuring[38–40], band sharpening[41], conduction band alignment[42] and Fermi level pinning[43]. These efforts have resulted in a maximum $ZT$ of ~1.7 at 850 K for PbS[40]. Nevertheless, because of the inferior TE performance at low temperature, for a long time, PbS was thought to be a medium-temperature thermoelectrics that can only operate at relatively high temperature. The investigation of its room

temperature TE performance has long been neglected[21], and the PbS had never been considered as a feasible TE cooling material[21,37–46].

In this work, we demonstrate that the thermoelectric performance of PbS can be significantly improved in the low temperature region, making it possible to work as a solid-state heat pump and could be a viable alternative to commercial $Bi_2Te_3$ material. As depicted in Fig. 1b, our optimal PbS sample exhibits a high power factor (PF), which is comparable to that of n-type commercial $Bi_2Te_3$ (COML BT), and possesses a higher $ZT$ than the COML BT above 373 K. Owing to the high TE performance and abundance in raw material[6,17], our PbS sample exhibits a higher $ZT_{ave}$ and much higher cost effectiveness (the ratio of $ZT_{ave}$ and cost) than the COML BT, as shown in Fig. 1c. The TE performance optimization route for PbS is shown in Fig. 1d, e. Here, we focus on improving the carrier mobility $\mu$ and low temperature TE performance of PbS through lattice plainification and Cu interstitial doping based on a PbS matrix with low thermal conductivity. Additional Pb was added to the matrix to realize lattice plainification, which compensates for the intrinsic cation vacancy and reduces the carrier scattering, achieving the simultaneous optimization of carrier mobility and carrier density. Furthermore, interstitial Cu was introduced to

denote free electron, optimizing the carrier density and improving the electrical transport properties over a wide temperature range. Through lattice plainification and interstitial doping, the optimized $Pb_{1.004}S_{0.6}Se_{0.4} + 0.001Cu$ polycrystal exhibits a decent room temperature $ZT$ of ~0.45. To further improve the carrier mobility and TE performance, the $Pb_{1.004}S_{0.6}Se_{0.4} + 0.001Cu$ compound was grown into high quality crystals (Fig. 1f). This results in a substantial increase in mobility and consequently enhances its PF to ~31.5 $\mu W\,cm^{-1}\,K^{-2}$ at room temperature. The promotion of electrical transport properties lead to the enhancement of $ZT$ in the whole investigated temperature range. Ultimately, the $Pb_{1.004}S_{0.6}Se_{0.4} + 0.001Cu$ crystal achieved a record high room temperature $ZT$ of ~0.64 and an average $ZT$ of ~0.92 in the temperature range of 300–823 K, see Fig. 1e. Based on this, the single-leg efficiency of our sample reached ~8% when subjected to a temperature difference of 474 K. (Fig. 1g) And a 7-pair TE module (shown in Fig. 1f) had been prepared by using the n-type $Pb_{1.004}S_{0.6}Se_{0.4} + 0.001Cu$ crystal and p-type commercial $Bi_2Te_3$ (Supplementary Fig. 1), which exhibits a cooling temperature difference of ~36.9 K when the hot end temperature at 303 K (Fig. 1g).

## Results

### Se alloying to suppress the thermal conductivity

Compared with PbSe and PbTe, PbS has a higher $\kappa_{lat}$, which limits the improvement of its $ZT$[17]. In order to optimize its thermal conductivity, we substituted part of S with Se and synthesized a series of $PbS_{1-x}Se_x$ by combining the high temperature melting with Spark Plasma Sintering (SPS). The powder X-ray diffraction (PXRD) results in Supplementary Fig. 2a shows that all $PbS_{1-x}Se_x$ ($x = 0$–0.5) samples are single phase and can be well indexed to the rock salt structure. Because of the larger atomic radius of Se than that of S, the diffraction peaks gradually shift to the low angle with increasing Se amount. The calculated lattice parameter in Supplementary Fig. 2b shows a linear dependence with increasing Se content, following the Vegard's law. This indicating the Se is successfully alloying into the PbS matrix.

Transport properties of $PbS_{1-x}Se_x$ ($x = 0$–0.5) samples were shown in Supplementary Figs. 3 and 4. The electrical transport properties of the materials are basically not affected by Se alloying (Supplementary Fig. 3a). However, heavy Se alloying makes the $\kappa_{lat}$ of the material greatly reduced due to the mass fluctuation and strain field fluctuation[37,47]. The room temperature $\kappa_{lat}$ reduces from 2.24 $W\,m^{-1}\,K^{-1}$ of PbS to 1.13 $W\,m^{-1}\,K^{-1}$ of $PbS_{0.6}Se_{0.4}$, representing a 50% reduction, and an extremely low $\kappa_{lat}$ of 0.82 $W\,m^{-1}\,K^{-1}$ was obtained in $PbS_{0.6}Se_{0.4}$ at 823 K, see Supplementary Fig. 3b. Comparing the $\kappa_{lat}$ of our $PbS_{0.6}Se_{0.4}$ sample with other reported lead chalcogenide materials, the $PbS_{0.6}Se_{0.4}$ shows a relatively low $\kappa_{lat}$, especially at room temperature[37,42–44,46–49] (Supplementary Fig. 3c). Finally, the highest room temperature $ZT$ of ~0.35 is obtained in $PbS_{0.6}Se_{0.4}$, as shown in Supplementary Fig. 3d, and its $ZT$ value improves in the whole investigated temperature range due to the decrease in $\kappa_{lat}$. Thus, the $PbS_{0.6}Se_{0.4}$ was chosen as the starting matrix for further optimization.

### Pb compensation for lattice plainification

It has been reported that fine tuning on eigenelements could contribute to the improvement in mobility and result in an enhanced $ZT$ value at room temperature both in theoretically and experimentally[48–51]. To optimize the electronic transport properties of the $PbS_{0.6}Se_{0.4}$ matrix, we conducted Pb compensation in this compound, and a series of polycrystalline $Pb_{1+y}S_{0.6}Se_{0.4}$ ($y = 0$–0.006) samples were synthesized. PXRD results in Supplementary Fig. 5a shows that all $Pb_{1+y}S_{0.6}Se_{0.4}$ samples are single phase. Lattice parameters of the samples are almost constant with increasing Pb, as shown in Supplementary Fig. 5b. The Scanning transmission electron microscopy (STEM) results of $PbS_{0.6}Se_{0.4}$ and $Pb_{1.004}S_{0.6}Se_{0.4}$ are shown in Fig. 2a, b. Obviously, some dark areas were observed in $PbS_{0.6}Se_{0.4}$, indicating the presence of intrinsic Pb vacancies in the

material. With adding extra Pb, these dark areas disappear in $Pb_{1.004}S_{0.6}Se_{0.4}$, implying the Pb vacancies were compensated. The line profile obtained from the zoom in area of $PbS_{0.6}Se_{0.4}$ and $Pb_{1.004}S_{0.6}Se_{0.4}$ is shown in Fig. 2c. The peak with weaker intensity indicated by the arrow corresponds to the presence of Pb vacancy. The densities of Pb vacancies in $PbS_{0.6}Se_{0.4}$ and $Pb_{1.004}S_{0.6}Se_{0.4}$ have also been directly counted from the STEM image, as shown in Supplementary Fig. 6. The density of Pb vacancy in $PbS_{0.6}Se_{0.4}$ is 0.28 $nm^{-2}$, and the number reduces to 0.03 $nm^{-2}$ after Pb compensation. These results support the conclusion that Pb compensation is an effective tool for reducing the intrinsic cation vacancy in $PbS_{0.6}Se_{0.4}$.

The electrical transport properties of $Pb_{1+y}S_{0.6}Se_{0.4}$ are demonstrated in Fig. 2d–g. By adding the extra Pb, the intrinsic cation vacancies were compensated, which reduces the carrier scattering brought by the cation vacancies. This improvement is reflected in the increase in $\mu$ from ~580 $cm^{-2}\,V^{-1}\,s^{-1}$ in the matrix to ~633 $cm^{-2}\,V^{-1}\,s^{-1}$ in $Pb_{1.002}S_{0.6}Se_{0.4}$, and it remains at ~614 $cm^{-2}\,V^{-1}\,s^{-1}$ in $Pb_{1.004}S_{0.6}Se_{0.4}$. In addition, the reduction in Pb vacancy also increases the electron density, from ~$2.32 \times 10^{18}\,cm^{-3}$ in the matrix to ~$3.71 \times 10^{18}\,cm^{-3}$ in $Pb_{1.004}S_{0.6}Se_{0.4}$, representing a 60% increase. (Fig. 2d) The realization of lattice plainification significantly improve the electrical conductivities of $Pb_{1+y}S_{0.6}Se_{0.4}$, particularly in the low temperature range. The highest room temperature $\sigma$ of 365 $S\,cm^{-1}$ was obtained in $Pb_{1.004}S_{0.6}Se_{0.4}$, representing a 70% enhancement compared to the pristine $PbS_{0.6}Se_{0.4}$. Although the increase in $n$ slightly lowers the $S$, the significant improvement in $\sigma$ result in the improved PF over a wide temperature range, see Fig. 2g. The average PF ($PF_{ave}$) at 300–823 K increased from ~7.7 $\mu W\,cm^{-1}\,K^{-2}$ in $PbS_{0.6}Se_{0.4}$ to ~10.4 $\mu W\,cm^{-1}\,K^{-2}$ in $Pb_{1.004}S_{0.6}Se_{0.4}$. The temperature dependent $\kappa_{tot}$ and $\kappa_{lat}$ of $Pb_{1+y}S_{0.6}Se_{0.4}$ are shown in Fig. 2h. Because of the reduction in cation vacancies and the phonon scattering, the $\kappa_{tot}$ and $\kappa_{lat}$ increases with adding extra Pb. The $\kappa_{lat}$ increases 10% in $Pb_{1.004}S_{0.6}Se_{0.4}$ compared with that of matrix. This in turn, supports the idea that the introduction of extra Pb is effective to cause the lattice plainification effect.

Although introducing the extra Pb reduces the intrinsic cation vacancy and enhances the $\kappa$, this effect also increases the $\mu$ and significantly improves the electrical transport properties. As a result, the TE performance has been improved over a wide temperature range, as shown in Fig. 2i. The $ZT_{ave}$ increased from ~0.36 in $PbS_{0.6}Se_{0.4}$ to ~0.47 in $Pb_{1.004}S_{0.6}Se_{0.4}$ in the temperature range of 300–823 K. The $Pb_{1.004}S_{0.6}Se_{0.4}$, exhibits highest PF and $ZT$, was chosen for further optimization.

### Interstitial Cu to optimize the carrier concentration

It has been proved that interstitial Cu plays an important role in wide temperature range $ZT$ enhancement for n-type lead chalcogenide[52–54]. Here, extra Cu was introduced to the $Pb_{1.004}S_{0.6}Se_{0.4}$ to optimize its electron density. PXRD result in Supplementary Fig. 8a shows that the diffraction peaks of all $Pb_{1.004}S_{0.6}Se_{0.4} + zCu$ ($z = 0$–0.006) samples can be indexed to the rock salt structure, and no obvious second phase has been observed. Because the additional Cu is in a tiny amount, the lattice parameters of all samples are almost constant, as shown in Supplementary Fig. 8b. The results of STEM demonstrate that Cu atoms form tiny cluster at nanoscale. (Fig. 3a) The amplified picture of Cu-rich area in Fig. 3b clearly shows that Cu atom occupy at interstitial site in the lattice. The line intensity scan profile shown in Supplementary Fig. 9a distinctly reveals the presence of Cu interstitial and prove it has an equal distance to the surrounding Pb/S atoms, indicating Cu occupies in the center of the cube made up of Pb and S/Se atoms. The multi-slice simulation of ADF-STEM image based on the crystal model is consistent with experimental results, as shown in Supplementary Fig. 9b.

To further reveal the formation mechanism of Cu interstitial, corresponding point defect formation energy was calculated. The details for the calculation method are shown in Supplementary

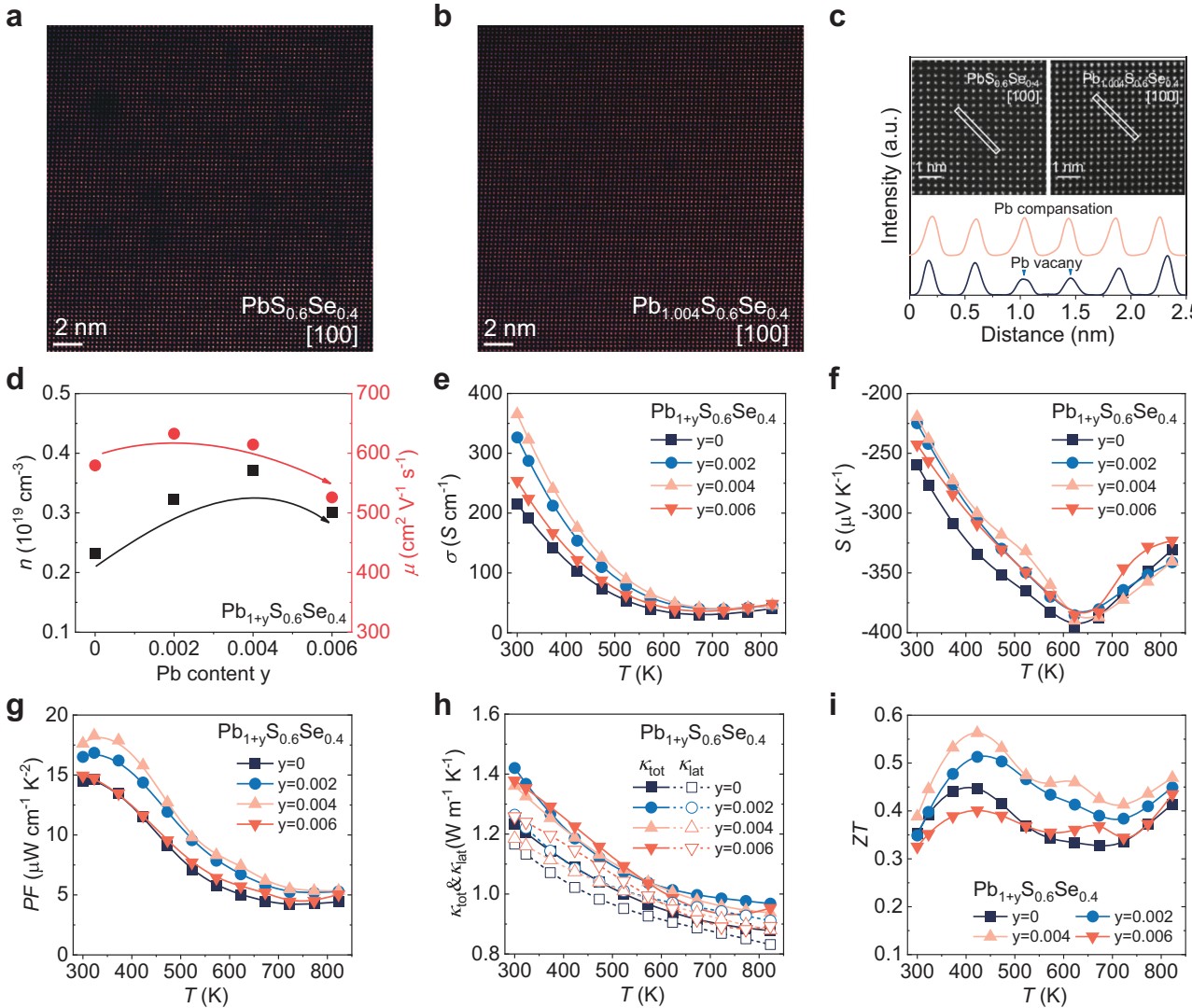

**Fig. 2 | Microstructure and transport properties of polycrystalline Pb$_{1+y}$S$_{0.6}$Se$_{0.4}$ samples.** ADF-STEM images of (**a**) PbS$_{0.6}$Se$_{0.4}$, (**b**) Pb$_{1.004}$S$_{0.6}$Se$_{0.4}$, respectively, both images along [100]; (**c**) line scan profiles obtained from enlarged images of PbS$_{0.6}$Se$_{0.4}$, and Pb$_{1.004}$S$_{0.6}$Se$_{0.4}$. **d** Room temperature carrier density $n$ and carrier mobility $\mu$ of Pb$_{1+y}$S$_{0.6}$Se$_{0.4}$ samples. Temperature dependence of (**e**) electrical conductivity $\sigma$, (**f**) Seebeck coefficient $S$, (**g**) power factor PF, (**h**) total thermal conductivity $\kappa_{tot}$ and lattice thermal conductivity $\kappa_{lat}$, (**i**) $ZT$ value of Pb$_{1+y}$S$_{0.6}$Se$_{0.4}$ samples.

information. Figure 3c and Supplementary Fig. 10 exhibit the point defect formation energy as a function of Fermi energy ($E_F$) in Pb-rich and S-rich conditions, respectively. Considering the Pb compensation carried out in our work, attention should be paid to the Pb-rich condition. As shown in Fig. 3c, when Cu enters the lattice interstitial, the system possesses the lowest formation energy. This indicates that Cu prefers to occupy the interstitial site in Pb-rich condition rather than replaces the Pb atom. Similar results has been observed in PbSe systems[53]. The defect formation energy calculation consolidates the STEM results, indicating the additional Cu atom occupied in the interstitial site, which donates free electron into the PbS matrix and optimizes the $n$. The calculated band gap results and optical band gap results show that the interstitial Cu has little impact on the band gap of Pb-S-Se system. Moreover, the adding Cu would make the Fermi level move toward the conduction band, as shown in Supplementary Fig. 11a–c.

Cu interstitial would denote free electrons to the matrix and the sample exhibits heavily doped degenerate semiconductor transport characteristics, as shown in Supplementary Fig. 11d and Supplementary Fig. 12. As Cu interstitial doping could optimize the $n$ of

Pb$_{1.004}$S$_{0.6}$Se$_{0.4}$ + zCu, the $\sigma$ was dramatically enhanced and the $S$ decreased gradually with increasing Cu content, as shown in Fig. 3d–f. The room temperature PF climbs up from ~17.6 $\mu$W cm$^{-1}$ K$^{-2}$ in Pb$_{1.004}$S$_{0.6}$Se$_{0.4}$ to ~20.6 $\mu$W cm$^{-1}$ K$^{-2}$ in Pb$_{1.004}$S$_{0.6}$Se$_{0.4}$ + 0.001Cu and then declines when further increasing the Cu content. Moreover, in the temperature range of 450 − 823 K, the PF of Pb$_{1.004}$S$_{0.6}$Se$_{0.4}$ + zCu had been significantly improved with adding Cu, which goes from ~5.3 $\mu$W cm$^{-1}$ K$^{-2}$ in Pb$_{1.004}$S$_{0.6}$Se$_{0.4}$ to ~13.2 $\mu$W cm$^{-1}$ K$^{-2}$ in Pb$_{1.004}$S$_{0.6}$Se$_{0.4}$ + 0.006Cu at 823 K (Fig. 3g). The interstitial Cu not only contribute to the enhancement of electrical transport properties, but also lead to the reduction in $\kappa_{lat}$ due to the enhanced phonon scattering. Figure 3h shows the thermal transport properties of Pb$_{1.004}$S$_{0.6}$Se$_{0.4}$ + zCu. Obviously, the effect of extra Cu on reducing $\kappa_{lat}$ become more pronounced as the temperature rises. We believe this is account for the dissolve of interstitial Cu, which has been previously reported[54]. The increased $\kappa_{tot}$ depicted in Fig. 3h is caused by the increased $\kappa_{ele}$ (Supplementary Fig. 13d). The simultaneous optimization of both electrical and thermal transport properties is acquired as a result of Cu interstitial doping, and the $ZT$ was enhanced in the whole investigated temperature range, see Fig. 3i. Among the Cu interstitial

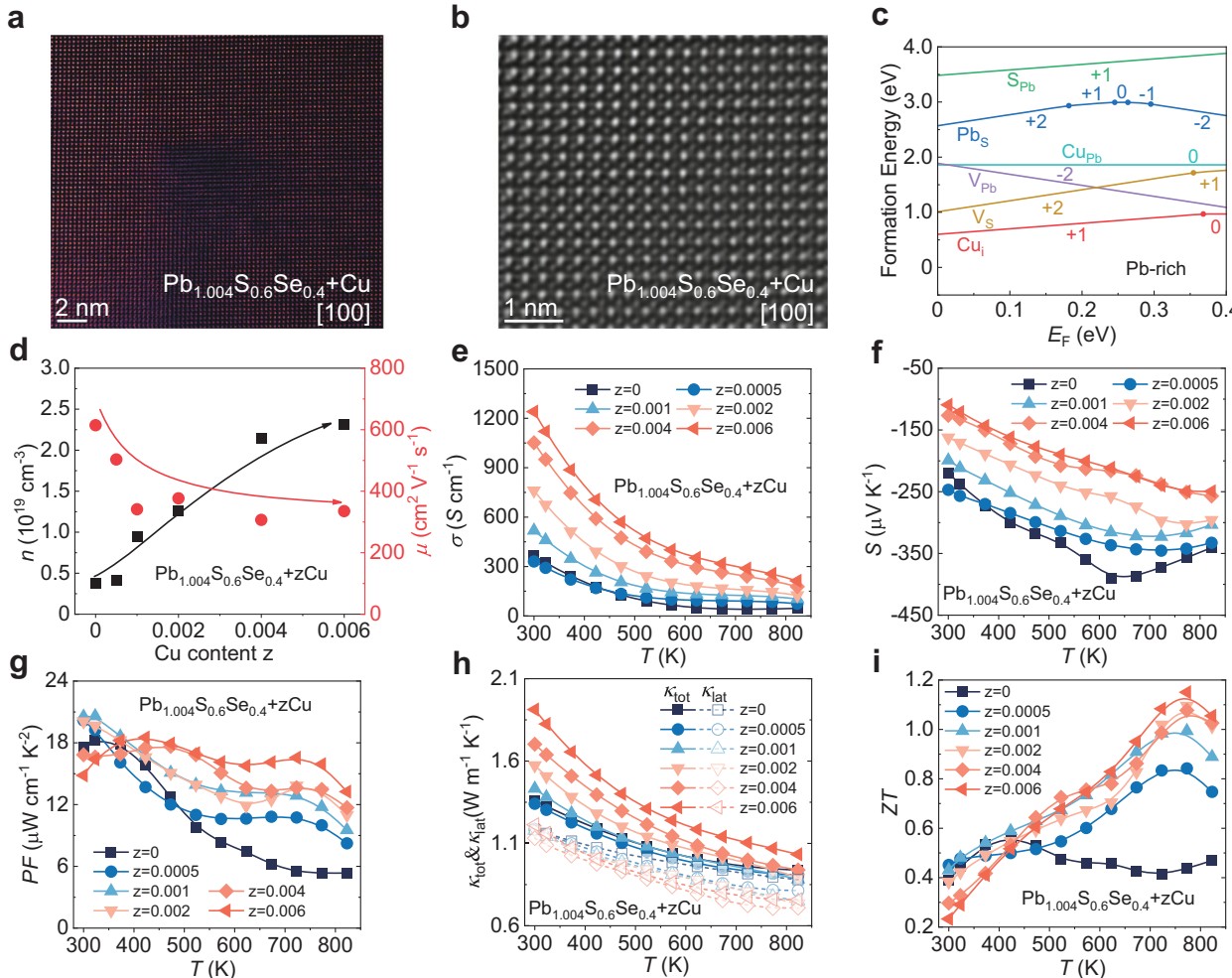

**Fig. 3 | Microstructure and transport properties of polycrystalline Pb$_{1.004}$S$_{0.6}$Se$_{0.4}$ + zCu samples. a** Atomic resolution STEM image with Cu-rich area along [100] zone axis and (**b**) amplifying image of Cu-rich area. **c** The formation energy calculation results for point defects in PbS as the function of Fermi energy ($E_F$) under Pb-rich condition. **d** Carrier density $n$ and carrier mobility $\mu$ of Pb$_{1.004}$S$_{0.6}$Se$_{0.4}$ + zCu samples. Temperature dependence of (**e**) electrical conductivity $\sigma$, (**f**) Seebeck coefficient $S$, (**g**) power factor PF, (**h**) total thermal conductivity $\kappa_{tot}$ and lattice thermal conductivity $\kappa_{lat}$, (**i**) $ZT$ value of Pb$_{1.004}$S$_{0.6}$Se$_{0.4}$ + zCu samples.

doped samples, Pb$_{1.004}$S$_{0.6}$Se$_{0.4}$ + 0.001Cu possesses the highest $ZT_{ave}$ of ~0.74 in the temperature range of 300–823 K, and it also obtained a peak $ZT$ value of ~1.0 at 773 K.

### Growing crystal to improve the carrier mobility and thermoelectric performance

To further enhance the electrical transport properties of Pb$_{1.004}$S$_{0.6}$Se$_{0.4}$ + 0.001Cu, we conducted crystal growth via vertical Bridgman method to reduce the carriers scattering on grain boundaries. The bulk XRD of the crystal cleavage plane for the crystal sample exhibits only the (200) and (400) diffraction peaks, in contrast to the polished SPS sample, which possesses the entire diffraction pattern of the rock salt structure. This implies the Pb$_{1.004}$S$_{0.6}$Se$_{0.4}$ + 0.001Cu crystal is well crystallized. Laue diffraction patterns in the insert of Fig. 4a and the picture of the crystal sample in Fig. 1f also confirmed the good quality of the crystal samples.

Scanning electron microscope (SEM) results exhibit the microstructure contrast between polycrystalline sample (Fig. 4b) obtained through SPS and crystal sample (Fig. 4c) obtained through vertical Bridgman method. In the SEM result of SPSed sample, obvious grain boundaries could be observed, the black spots are the holes formed during the sintering process. On the contrary, the crystal sample shows no grain boundary at the same magnification scale. Figure 4d shows

the significant improvement of $\mu$ in the crystal sample compared with the SPSed sample. The $\mu$ reaches ~552 cm$^{-2}$ V$^{-1}$ s$^{-1}$ in crystal sample and realizes ~60% increase. This high $\mu$ in crystal sample is caused by the reduction of carrier scattering on grain boundary.

On account of the optimized $\mu$, the crystal sample shows extraordinary TE performance compared with the SPSed sample. Figure 4e–g exhibits the electrical transport properties of SPSed sample and crystal sample. The crystal sample has a higher $\sigma$, this increase is caused by the improved $\mu$, which resulted from the reduction of carriers scattering on the grain boundaries. In addition, the lower $n$ leads to the enhancement in $S$, thus making the crystal sample exhibit a higher PF, especially in low temperature region, the crystal obtained a peak PF of ~31.5 μW cm$^{-1}$ K$^{-2}$ at room temperature. The crystal sample also reduces the phonon scattering on the grain boundaries, so the $\kappa_{lat}$ of crystal sample is a slightly higher than that of the SPS sample, as shown in Fig. 4h. Owing to the elevated PF, the $ZT$ of crystal sample achieves improvement over a wide temperature range. As shown in Fig. 3i, the crystal sample obtains a room temperature $ZT$ of ~0.64 and a peak $ZT$ of ~1.1 at 823 K.

As depicted in Fig. 5a, compared with other PbS-based materials, our sample has a higher weighted mobility $\mu_W$, especially in low temperature range. Due to the high $\mu_W$, our sample shows a significant advantage in PF below 523 K, as shown in Fig. 5b. The PF$_{ave}$ between

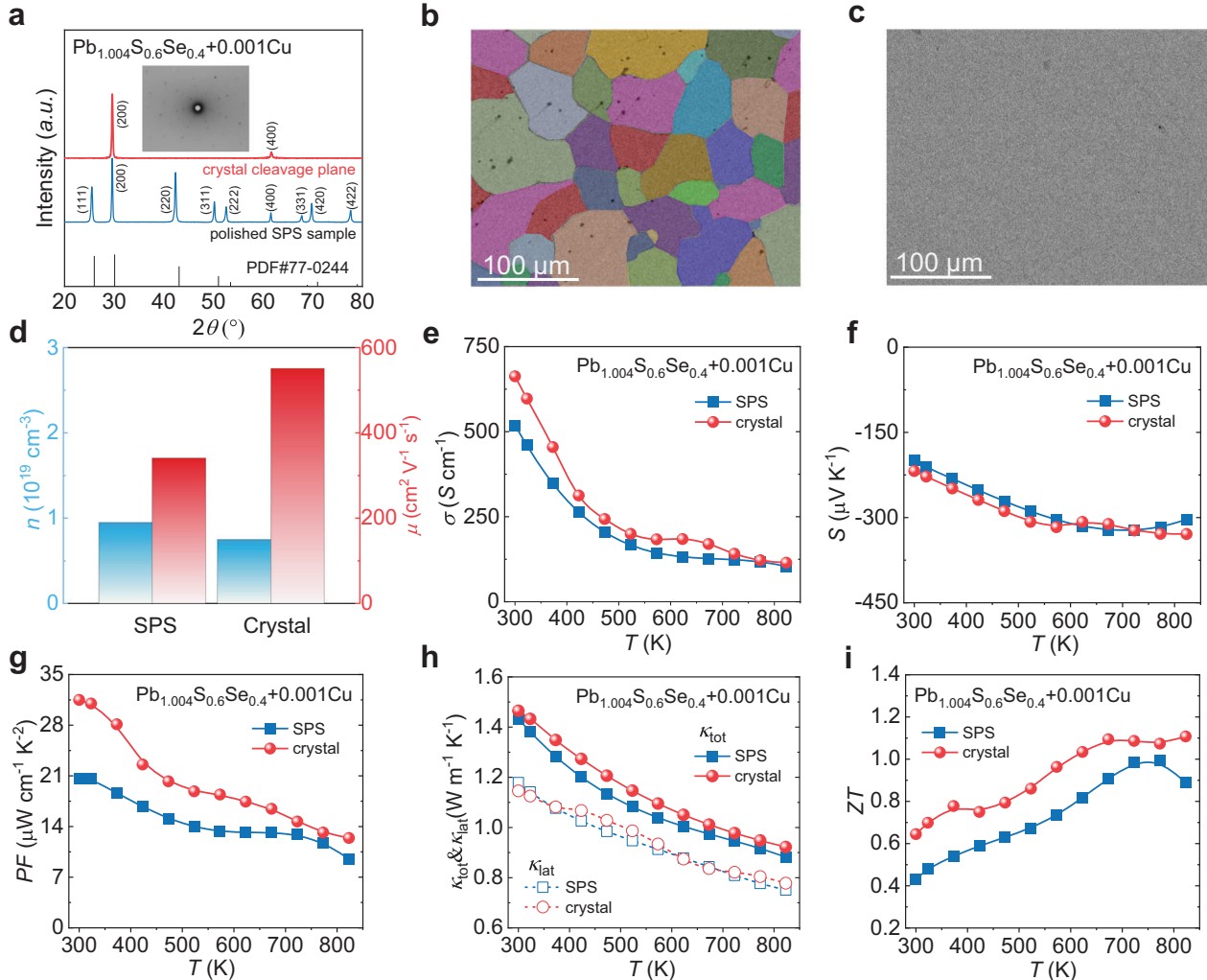

**Fig. 4 | Microstructure and transport properties of SPSed and crystalline Pb$_{1.004}$S$_{0.6}$Se$_{0.4}$ + 0.001Cu samples. a** XRD results measured on a polished SPSed sample and the cleavage plane of crystal, inset figure shows Laue pattern of the crystal sample. Microstructure comparison between (**b**) SPSed sample and (**c**) crystal sample. **d** Comparison of room temperature carrier density $n$ and carrier mobility $\mu$ between SPSed sample and crystal sample. Comparison of the temperature dependence of (**e**) electrical conductivity $\sigma$, (**f**) Seebeck coefficient $S$, (**g**) power factor PF, (**h**) total thermal conductivity $\kappa_{tot}$ and lattice thermal conductivity $\kappa_{lat}$, (**i**) ZT value of Pb$_{1.004}$S$_{0.6}$Se$_{0.4}$ + 0.001Cu between SPSed and ingot samples.

300 and 523 K reaches ~24.7 μW cm$^{-1}$ K$^{-2}$, which is a fairly high value for the PbS-based compound. (Fig. 5c) The value of $\mu_W/\kappa_{lat}$ indicates that the low $\kappa_{lat}$ brought by Se alloying and Cu interstitial doping also matters for the elevated ZT compared with other PbS-based materials, as shown in Fig. 5d, e. Finally, because of the improvement in TE performance at room temperature, the ZT$_{ave}$ of our sample reaches ~0.92 at the temperature range of 300–823 K, as depicted in Fig. 5f, which is the highest ZT$_{ave}$ among PbS-based material systems[21,37–46].

To further evaluate the TE performance of the Pb$_{1.004}$S$_{0.6}$Se$_{0.4}$ + 0.001Cu crystal, we built a 7-pair TE module based on our n-type material and commercial p-type Bi$_2$Te$_3$. The maximum cooling temperature difference of this module reached ~36.9 K at the hot end temperature of 303 K, (Supplementary Fig. 14) which is close to the reported maximum cooling temperature difference of ~42.6 K for PbTe-based device[49], indicating that PbS also has the potential to serve as a TE cooling material. The high ZT$_{ave}$ of ~0.92 shown in the crystal sample also implies that our work has an advantage in power generation. Thus, we conducted a single-leg energy conversion efficiency test to evaluate its performance. Supplementary Fig. 15 and Fig. 5g, h show the experiment data of single-leg power generation with the $T_c$ of 295 K, including output voltage $U$, output power $P$ and conversion efficiency $\eta$ as a function of the electric current $I$. The y-axis value in Supplementary Fig. 15 shows an open-circuit voltage that relates to the Seebeck voltage of the single-leg device, which rises from ~5 mV at a $\Delta T$ of 33 K to ~110 mV at a $\Delta T$ of 474 K. The $P$–$I$ plot in Fig. 5g indicates the single-leg device possesses a maximum $P$ of ~38 mV. The power generation capacity increases with the elevated $\Delta T$ owing to the continuously increasing ZT$_{ave}$ of the crystal sample in the investigated temperature range, and it achieves a high efficiency of ~8% when $I$ = 0.7 A, $\Delta T$ = 474 K (Fig. 5h). Figure 5i shows a comparison of the TE power generation efficiency of our sample with other TE materials. Compared with other sulfur-based TE materials[39,55–58], our work shows outstanding performance and achieves a high efficiency at a temperature difference of 474 K.

## Discussion

More than 60% of energy generated by fossil fuels is dissipated as waste heat, which is challenging to directly recover and utilize[1]. Thermoelectric materials, by converting heat directly into electrical energy, enable the recovery and utilization of low-grade waste heat resources, thereby improving the energy utilization and sustainability, reducing the energy costs and carbon emissions. Additionally, thermoelectric cooling technology is considered an ideal choice for temperature control of electronic devices due to its high reliability, rapid response,

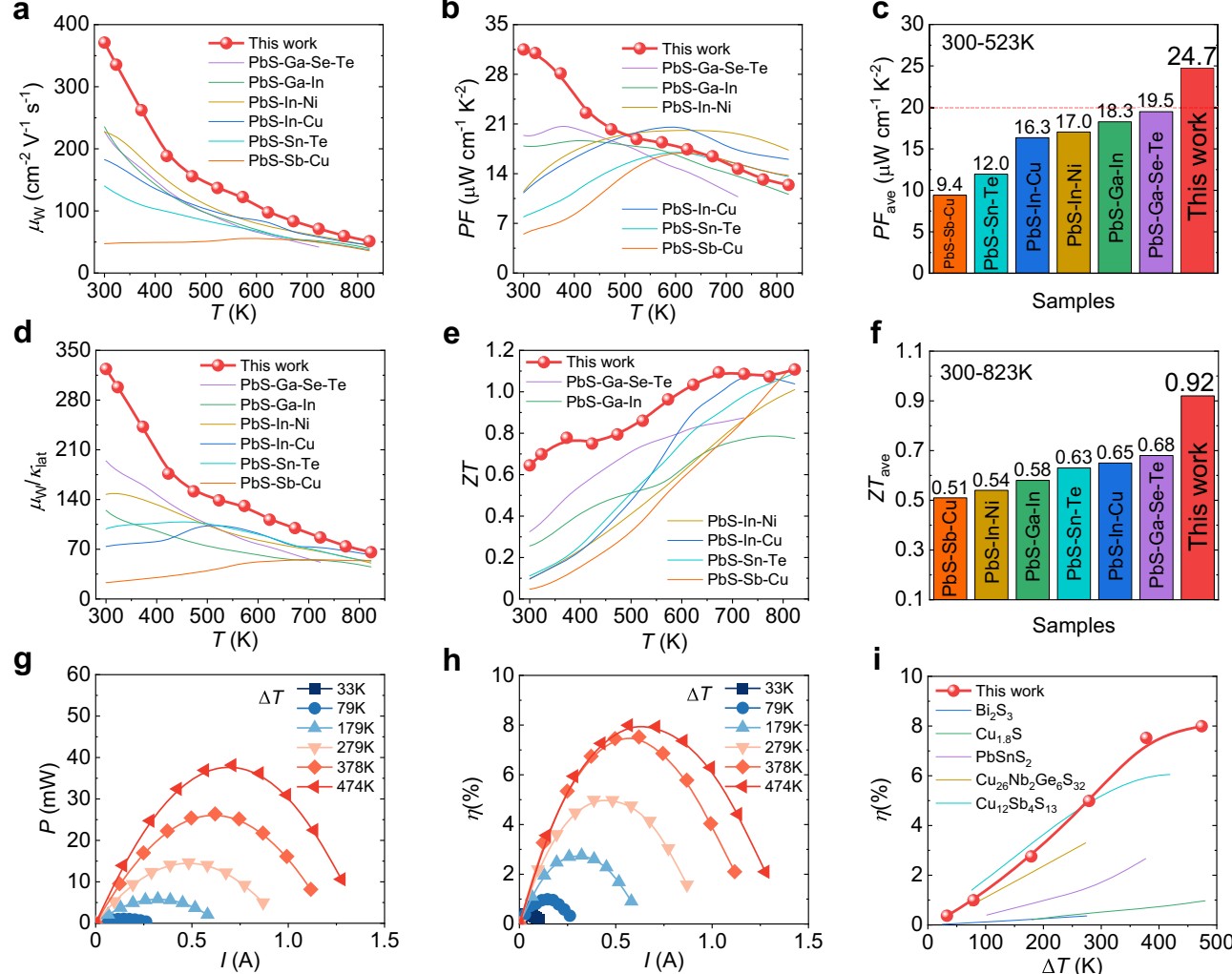

**Fig. 5 | Device performance of the crystalline Pb$_{1.004}$S$_{0.6}$Se$_{0.4}$ + 0.001Cu sample.** Comparison of the **a** weighted mobility $\mu_W$, **b** power factor PF, **c** average power factor PF$_{ave}$, **d** ratio of $\mu_W$ and lattice thermal conductivity, **e** ZT value, **f** average ZT of this work and other PbS-based materials. The experimentally measured **g** output power P and **h** conversion efficiency $\eta$ with respect to electric current I of the Pb$_{1.004}$Se$_{0.6}$S$_{0.4}$ + 0.001Cu crystal sample, **i** Comparison of the power generation efficiency $\eta$ for our sample and other sulfur-based TE materials.

precise temperature control, and ease of miniaturization. Despite the promising applications of thermoelectric technology, existing commercial thermoelectric material contain the rare element tellurium, hindering the sustainable and widespread application of thermoelectric technology.

In this study, we demonstrate the low-cost and earth-abundant PbS holds great potential for use in both thermoelectric cooling and power generation. Lattice plainification and interstitial doping strategies were employed to optimize the electronic transport properties of PbS, and a remarkable power factor of ~31.5 $\mu$W cm$^{-1}$ K$^{-2}$ and a record high room temperature ZT of ~0.64 have been achieved in the Pb$_{1.004}$S$_{0.6}$Se$_{0.4}$−0.001Cu crystal. Based on this, the first thermoelectric cooling module based on n-type PbS was fabricated, gaining a maximum temperature difference of ~36.9 K when the hot end temperature is 303 K. Furthermore, a single-leg device using our PbS material demonstrates a relatively high power generation efficiency of ~8% at a temperature difference of 474 K, indicating that PbS holds significant potential for harvesting waste heat into valuable electrical power. This study presents a systematic approach to optimizing the low temperature thermoelectric performance of PbS, and demonstrates that PbS could be a viable alternative to commercial Bi$_2$Te$_3$. This development contributes to the sustainability of thermoelectric technology.

## Methods

### Sample synthesis
All samples were weighted with nominal ratio using high-purity raw material, put in quartz tubes and flame-sealed after vacuum. They were heated to 723 K in 12 h, then took 7 h from 723 K to 1403 K, kept at 1403 K for 6 h, finally the polycrystalline ingots were obtained after furnace cooling. Compact polycrystalline samples were obtained through Spark Plasma Sintering (SPS-211Lx) at 873 K for 6 min under 50 MPa compressive stress using the powder of the polycrystalline ingots. The crystal samples were obtained through the vertical Bridgman method, they were first synthesized with the same procedure as polycrystal, then the obtained ingot was grounded into powder. The powder was loaded into silica tubes and flame sealed at a residual pressure under 10$^{-4}$ Pa. Then it was heated to 1403 K in 11 h, kept for 3 h and cooled at a rate of 1 K per h from 1403 K to 1303 K. Then the furnace was shut down and cooled to room temperature.

### Structural characterization
The PXRD patterns of all samples were obtained using D/max2200PC instrument operating at 40 KV and 40 mA with Cu K$\alpha$ ($\lambda$ = 1.5418 Å) radiation. The Laue pattern was obtained on a diffractometer operating at 20 KV and 20 mA.

## Microstructure investigation

The Scanning electron microscopy (SEM) was carried out through a field emission SEM (JEOL, JSM-7000F) with an acceleration voltage of 20 KV. The High-resolution Scanning transmission electron microscopy (HR-STEM) was carried out through JEM F200 with a Schottky hot field gun at 200 KV. The high-resolution images in Figs. 2c and 3b and Supplementary Fig. 9a were Gaussian blurred for better visualization.

## Hall measurements

Hall coefficient ($R_H$) of all samples with thickness of ~0.7 mm was measured through Lake Shore 8400 Series with invertible magnetic field of 0.9 T and current of 15 mA applied. The carrier density ($n_H$) and carrier mobility ($\mu_H$) were calculated through $n_H = 1/(eR_H)$, $\mu_H = \sigma R_H$, respectively.

## Thermoelectric transport properties measurements

All samples were cut into cuboids with sample size of ~10 × 3 × 3 mm$^3$ and slices with sample size of ~8 × 8 × 1.5 mm$^3$ for electrical and thermal properties measurements, respectively. The Seebeck coefficient $S$ and electrical conductivity $\sigma$ were obtained concurrently through Cryoall CTA instrument under a low-pressure helium atmosphere. Total thermal conductivity $\kappa_{tot}$ was calculated through $\kappa_{tot} = D\rho C_p$. Thermal diffusivity $D$ was obtained through Cryoall CLA1000 and analyzed using Cowan model with pulse correction. Sample density $\rho$ was calculated through sample mass and dimensions. The specific heat capacity $C_p$ was calculated through Debye model. The combined uncertainty for all measurements involved in the calculation of $ZT$ was less than 20%.

## Single leg conversion efficiency test

The single leg power generation efficiency is conducted by Mini-PEM instrument (Advance Riko) under vacuum, using a nominal hot-side temperature at 300–773 K. The hot and cold sides of the single-leg were galvanized to form Ni-based barrier layer using commercial nickel-plating solution, and copper wires were soldered by Ag-based solder.

## Thermoelectric module fabrication and cooling performance test

The optimized PbS ingot and commercial p-type Bi$_2$Te$_3$ were cut into cuboids with size of 2 × 2 × 4 mm$^3$ and then put into ethyl alcohol for ultrasonic cleaning. Then the cuboids were galvanized to form Ni-based barrier layer using commercial nickel-plating solution, and then the cuboids were welded on Cu electrode using Sn$_{48}$Bi$_{52}$ solder paste. Finally, the n-type PbS-based thermoelectric cooling module was fabricated and the cooling performance was test using Z-meter DX4090.

## Data availability

The authors declare that the data supporting the findings of this study are available on reasonable request.

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

## Acknowledgements

This work was primary supported by the National Science Fund for Distinguished Young Scholars (51925101), National Natural Science Foundation of China (52250090, 52371208, 52002042, 51772012, 51571007 and 12374023), the Beijing Natural Science Foundation (JQ18004), and the 111 Project (B17002). L.-D.Z. thanks for the support from Tencent Xplorer Prize.

## Author contributions

The manuscript was written through the contributions of all authors. All authors have approved the results and conclusions of this work. Lei Wang: investigation, data curation, writing, original draft; Yi Wen: conducted the SEM and STEM experiments; Shulin Bai: theoretical calculation; Cheng Chang, Shan Liu, Dongrui Liu, Siqi Wang, Zhe Zhao, Shaoping Zhan, Qian Cao, Xiang Gao: data analysis; Yichen Li: validation; Hongyao Xie and Li-Dong Zhao: research design, data analysis, draft writing and editing, and funding acquisition.

## Competing interests

The authors declare no competing interests.
