## [Peer Review File · Nature Communications]

Realizing Thermoelectric Cooling and Power Generation in N-type $\text{PbS}_{0.6}\text{Se}_{0.4}$ via Lattice Plainification and Interstitial DopingREVIEWER COMMENTS

Reviewer #1 (Remarks to the Author):

The paper by Wang et al. is devoted to the study of thermoelectric (TE) properties of some lead chalcogenides. Starting with the ternary $\text{PbSe}_{1-x}\text{S}_x$ solid solution and doping it with the smallest amount of copper authors manage to achieve the TE efficiency which is comparable with those of the state-of-the-art materials. Furthermore, $\text{PbSe}_{1-x}\text{S}_x$ is for sure much cheaper than Te-containing compounds. The paper is nicely written and well organized. It is also an interesting contribution to the solid state physics. However, I cannot recommend this work for publication in Nature Communication, since the concept of the “lattice planification” is not innovative and was already shown by the same authors (see DOI: 10.1126/science.adg7196) to be a powerful tool in improvement of TE efficiency. Also, ZT-values of 1-2 are known for a number of lead chalcogenides and obviously preparation of a complex doped $\text{PbSe}_{1-x}\text{S}_x$ (despite being cheaper than Bi_2Te_3) would be rather a challenge for industry. After some improvement this work should be resubmitted to e.g., Chemistry of Materials.

1) Title of the paper should be for sure changed. Now it sounds like authors would take natural galena, slightly modify it and immediately obtain high TE efficiency. And indeed, they are investigating synthetic Pb-Se-S solid solutions, which do not occur in the nature. Provide in the title true composition of your material.

2) Author are writing that “intrinsic cation vacancies were compensated by adding the extra Pb” (raw 144). However, no data justifying such a statement are provided. Obviously, some experiments estimating vacancies concentrations before and after doping should be performed. Corresponding numbers should be presented and discussed.

3) In the title and in the further text author are writing about “lattice planification effect”, which can be caused by adding Pb. Please provide more details on this issue and “quantify” this effect, giving some numbers, relations etc.

4) The role of “Cu-cluster” in the enhancement of TE-performance of $\text{PbSe}_{1-x}\text{S}_x$ is totally unclear. First, what does it mean “Cu occupy at interstitial site”: which one (i.e, Wyckoff position)? Provide a clear crystallographic description with coordination polyhedra for Cu-atoms as well as the interatomic distances therein. Obviously, to answer these questions an additional TEM study would be required.

5) In the discussion to Figs. 2 and 3 authors are speculating on the temperature evolutions of electrical resistivity, thermal conductivity and Seebeck coefficient of the studied materials basing on the room temperature values for charge carrier concentrations and their mobilities. Obviously, to make such discussions $n(T)$ and $\mu(T)$ should be measured as well.

6) The theoretical calculations part should be extended. What happens to the energy gap with increasing Cu-content in Pb-Se-S? Show some electronic density of state, compare theoretically calculated E_g -values with those deduced from the Arrhenius plots to your $\sigma(T)$ data. The trends from theory and experiment are expected to agree with each other.

7) Fifty citations are obviously insufficient to reflect the state-of-the-art in the TE properties of lead chalcogenides. Authors have to analyze more carefully the literature data.

Reviewer #2 (Remarks to the Author):

In this work, lattice planification and interstitial doping strategies were employed to optimize the thermoelectric performance of n-type PbS crystal, resulting in a remarkable power generation efficiency of ~8% in the material. Additionally, this work demonstrates the first thermoelectric cooling device based on the n-type PbS material, which exhibits a decent cooling temperature difference of ~37 K at room temperature. This study shows the low-cost and earth-abundant PbS crystal could be a viable alternative to commercial n-type Bi₂Te₃ thermoelectric materials. I believe this work is sound and comprises substantial novelty and significance. Therefore, it is important to publish the present results as they provide new insights to the PbS thermoelectric material. I recommend this interesting work for publication after a minor revision.

1. I did not find the fabrication process of the TE device in the experimental section. Please provide the details.
2. When comparing the TE performance of their PbS sample with other reported materials, the authors calculated the average ZT from 300 to 823 K, but for the average power factor, they only considered values from 300 to 523 K. Is there a specific reason for this difference? It should be explained.
3. Why the Cu clusters look darker in the STEM image?
4. There are few typos in the figure. In Fig.4h, the crystal sample has been labeled as “ingot 2”, it should be changed to “crystal”; In the legend of Fig 2f, the Pb content was label as “x”, while in other panels, it was named as “y”, please correct.

Reviewer #3 (Remarks to the Author):

attached

Comment:

In this work, the authors prepared a series of $\text{Pb}_{1.004}\text{S}_{0.6}\text{Se}_{0.4+\text{Cu}}$ samples and significantly improved their thermoelectric performance by employing the lattice planification and interstitial doping strategies. They found that, minor Pb compensation improves the crystal quality and enhances carrier mobility, while interstitial effectively increases the carrier density. These effects result in a high conversion efficiency of $\sim 8\%$ in the material. Furthermore, this work demonstrates the first PbS-based thermoelectric cooling device, which exhibits an impressive cooling temperature difference of ~ 36.9 at room temperature. All the data are analyzed properly, and the paper is well written. Thereby, I recommend this work for publication with minor revision. Additionally, I have a few suggestions which author should consider during revision:

1. The 0.6% Cu-doped sample exhibits the highest TE performance. What would happen if further increases in Cu content? Besides, the authors should clarify why they selected the 0.1% Cu-doped sample for device fabrication.

2. More details about the fabrication process of the TE device, including steps such as material preparation, device assembly, and soldering procedure, should be provided in the methods section.

3. Why the lattice parameter did not change when adding extra Cu into the lattice. The authors should provide an explanation for this observation.

4. More details about the crystal preparation process should be provided, particularly regarding what happens after cooling the crystal sample at a rate of 1 K per h from 1403 K to 1303 K.

5. The authors should correct any typos found in the paper.

Reviewer(s)' Comments to Author:

Reviewer: 1

The paper by Wang et al. is devoted to the study of thermoelectric (TE) properties of some lead chalcogenides. Starting with the ternary $\text{PbSe}_x\text{S}_{1-x}$ solid solution and doping it with the smallest amount of copper authors manage to achieve the TE efficiency which is comparable with those of the state-of-the-art materials. Furthermore, $\text{PbSe}_x\text{S}_{1-x}$ is for sure much cheaper than Te-containing compounds. The paper is nicely written and well organized. It is also an interesting contribution to the solid state physics. However, I cannot recommend this work for publication in Nature Communication, since the concept of the “lattice plainification” is not innovative and was already shown by the same authors (see DOI: 10.1126/science.adg7196) to be a powerful tool in improvement of TE efficiency. Also, ZT-values of 1-2 are known for a number of lead chalcogenides and obviously preparation of a complex doped $\text{PbSe}_x\text{S}_{1-x}$ (despite being cheaper than Bi_2Te_3) would be rather a challenge for industry. After some improvement this work should be resubmitted to e.g., Chemistry of Materials.

Reply: Thanks for your comments, previous studies on lattice plainification were conducted in SnSe materials, while this work focuses on $\text{PbS}_{0.6}\text{Se}_{0.4}$ materials. Optimizing the thermoelectric properties of $\text{PbS}_{0.6}\text{Se}_{0.4}$ materials through lattice plainification strategies has not been reported before. In addition, the core innovation of this work does not emphasize the lattice plainification strategies. Instead, our main innovation lies in demonstrating the room temperature refrigeration performance in a conventional high temperature TE material $\text{PbS}_{0.6}\text{Se}_{0.4}$, for the first time. For a long time PbS and its derivatives have been considered good TE materials at high temperature, the highest ZT value has been widely extended while the research on near room temperature and refrigeration performance has been neglected. This work boosts the room temperature thermoelectric performance of $\text{PbS}_{0.6}\text{Se}_{0.4}$ and reports the first n-type $\text{PbS}_{0.6}\text{Se}_{0.4}$ -based TE cooling device, demonstrating that sulfur-based materials also have the potential for refrigeration. Furthermore, the chemical composition of the $\text{PbS}_{0.6}\text{Se}_{0.4}$ -Cu material we developed is not more complex than the current commercial thermoelectric materials. Currently, the common commercial n-type thermoelectric material is Iodine-doped $\text{Bi}_2\text{Te}_{2.7}\text{Se}_{0.3}$, which is also a kind of four-component compound. Moreover, our material can be prepared by the Bridgman method, which is widely used in the manufacture of commercial TE material. Thus, we don't think the preparation of our material would pose a challenge for industry.

Question/Comments 1: Title of the paper should be for sure changed. Now it sounds like authors would take natural galena, slightly modify it and immediately obtain high TE efficiency. And indeed, they are investigating synthetic Pb-Se-S solid solutions, which do not occur in the nature. Provide in the title true composition of your material.

Reply: Thanks for your comments, the title has been changed to “Realizing Thermoelectric Cooling and Power Generation in N-type $\text{PbS}_{0.6}\text{Se}_{0.4}$ via Lattice Plainification and Interstitial Doping”

Question/Comments 2: Author are writing that “intrinsic cation vacancies were compensated by adding the extra Pb” (raw 144). However, no data justifying such a statement are provided. Obviously, some experiments estimating vacancies concentrations before and after doping

should be performed. Corresponding numbers should be presented and discussed.

Reply: Thanks for your reminding. We have added some STEM results of $\text{PbS}_{0.6}\text{Se}_{0.4}$ and $\text{Pb}_{1.004}\text{S}_{0.6}\text{Se}_{0.4}$ to support this statement. As shown in Fig. R1(a) and (b), some dark areas were observed in $\text{PbS}_{0.6}\text{Se}_{0.4}$, indicating the presence of intrinsic Pb vacancies in the pristine $\text{PbS}_{0.6}\text{Se}_{0.4}$. With adding extra Pb, these dark areas disappear in $\text{Pb}_{1.004}\text{S}_{0.6}\text{Se}_{0.4}$, implying the Pb vacancies were compensated. The line profile obtained from the zoom in area of $\text{PbS}_{0.6}\text{Se}_{0.4}$ and $\text{Pb}_{1.004}\text{S}_{0.6}\text{Se}_{0.4}$ is shown in Fig. R1(c) and (d), respectively. The peak with weaker intensity indicated by the arrow corresponds to the presence of Pb vacancies. The density of Pb vacancy in $\text{PbS}_{0.6}\text{Se}_{0.4}$ and $\text{Pb}_{1.004}\text{S}_{0.6}\text{Se}_{0.4}$ have also been directly counted from the STEM image, as shown in Fig. R1(e) and (f). The density of Pb vacancy in $\text{PbS}_{0.6}\text{Se}_{0.4}$ is 0.28 nm^{-2} , and this number reduces to 0.03 nm^{-2} after Pb compensation. These data strongly support the Pb compensation is an effective tool to reduce the intrinsic cation vacancy in $\text{PbS}_{0.6}\text{Se}_{0.4}$.

The corresponding data and discussion had been added into the revised manuscript.

Figure R1. ADF-STEM images of (a) $\text{PbS}_{0.6}\text{Se}_{0.4}$, and (b) $\text{Pb}_{1.004}\text{S}_{0.6}\text{Se}_{0.4}$, respectively, both

imaged along [100]; enlarged images and line intensity scan profiles showing (c) Pb vacancy and (d) Pb compensation; counting of vacancies in (e) $\text{PbS}_{0.6}\text{Se}_{0.4}$ and (f) $\text{Pb}_{1.004}\text{S}_{0.6}\text{Se}_{0.4}$, respectively, both imaged along [100]. Before Pb compensation, the density of vacancy was 0.28 nm^{-2} . After Pb compensation, the density of vacancy reduces to 0.03 nm^{-2} .

Question/Comments 3: In the title and in the further text author are writing about “lattice planification effect”, which can be caused by adding Pb. Please provide more details on this issue and “quantify” this effect, giving some numbers, relations etc.

Reply: The lattice planification effect of adding Pb is reflected in the increase in μ from $\sim 580 \text{ cm}^{-2} \text{ V}^{-1} \text{ s}^{-1}$ in the matrix to $633 \text{ cm}^{-2} \text{ V}^{-1} \text{ s}^{-1}$ in $\text{Pb}_{1.002}\text{S}_{0.6}\text{Se}_{0.4}$, and it maintains at $\sim 614 \text{ cm}^{-2} \text{ V}^{-1} \text{ s}^{-1}$ in $\text{Pb}_{1.004}\text{S}_{0.6}\text{Se}_{0.4}$. In addition, the reduction in Pb vacancy also increases the electron density, from $\sim 2.32 \times 10^{18} \text{ cm}^{-3}$ in the matrix to $\sim 3.71 \times 10^{18} \text{ cm}^{-3}$ in $\text{Pb}_{1.004}\text{S}_{0.6}\text{Se}_{0.4}$, representing a 60% increase. The highest room temperature σ of 365 S cm^{-1} was obtained in $\text{Pb}_{1.004}\text{S}_{0.6}\text{Se}_{0.4}$, representing a 70% enhancement compared to the pristine $\text{PbS}_{0.6}\text{Se}_{0.4}$. The lattice thermal conductivity also increases 10% in $\text{Pb}_{1.004}\text{S}_{0.6}\text{Se}_{0.4}$ compared with that of matrix due to the reduction of phonon scattering caused by the Pb vacancy. All these effects have been discussed in the manuscript.

Question/Comments 4: The role of “Cu-cluster” in the enhancement of TE-performance of $\text{PbSe}_x\text{S}_{1-x}$ is totally unclear. First, what does it mean “Cu occupy at interstitial site”: which one (i.e, Wyckoff position)? Provide a clear crystallographic description with coordination polyhedra for Cu-atoms as well as the interatomic distances therein. Obviously, to answer these questions an additional TEM study would be required.

Reply: Thanks for your comments. We have added clearer STEM results to better reveal the occupancy of Cu atoms. Fig. R2a shows the ADF-STEM images of the $\text{Pb}_{1.004}\text{S}_{0.6}\text{Se}_{0.4}+0.001\text{Cu}$ sample along the [100] axis, the Cu-rich area appears darker due to the diffraction contrast. Cu interstitial can be seen clearly in the high-resolution image shown in Fig. R2b. The line intensity scan profile shown in Fig. R2c distinctly reveals the presence of Cu interstitial. The distance between two Pb/S peak centers obtained from the STEM image is 0.42 nm , which agrees with the theoretical value. The distance between Pb/S peak center and Cu peak center is 0.21 nm , indicating Cu occupies the center of the cube made up of Pb and S atoms. Fig. R2d demonstrates multislice simulation of ADF-STEM based on crystal model showing Cu interstitial atoms in PbS matrix, and the obtained simulated results are consistent with experimental STEM images.

The corresponding data and discussion had been added into the revised manuscript.

Figure R2. (a) ADF-STEM images and (b) high resolution images of $\text{Pb}_{1.004}\text{S}_{0.6}\text{Se}_{0.4} + 0.001\text{Cu}$ sample along [100]; (c) line intensity scan profile of $\text{Pb}_{1.004}\text{S}_{0.6}\text{Se}_{0.4} + 0.001\text{Cu}$ sample; (d) multislice simulation of ADF-STEM image based on crystal model showing Cu interstitial atoms (blue spheres) in PbS matrix (Pb - gray spheres, S - yellow spheres).

Question/Comments 5: In the discussion to Figs. 2 and 3 authors are speculating on the temperature evolutions of electrical resistivity, thermal conductivity and Seebeck coefficient of the studied materials basing on the room temperature values for charge carrier concentrations and their mobilities. Obviously, to make such discussions $n(T)$ and $\mu(T)$ should be measured as well.

Reply: Thanks for your suggestion. We have conducted the $n(T)$ and $\mu(T)$ measurement in the temperature range of 300 - 523 K for $\text{Pb}_{1.004}\text{S}_{0.6}\text{Se}_{0.4}$ and $\text{Pb}_{1.004}\text{S}_{0.6}\text{Se}_{0.4} + 0.001\text{Cu}$, the results are shown in Fig R3. The carrier concentration increases with adding Cu, since the interstitial Cu is able to donate free electron into matrix. Additionally, the carrier concentration of all samples does not change with increasing temperature, and their carrier mobility significantly decreases with rising temperature, exhibiting the degenerated semiconductor behavior.

The corresponding data and discussion had been added into the revised supporting information.

Figure R3 The temperature dependent (a) carrier concentration and (b) carrier mobility of $\text{Pb}_{1.004}\text{S}_{0.6}\text{Se}_{0.4}$ and $\text{Pb}_{1.004}\text{S}_{0.6}\text{Se}_{0.4} + 0.001\text{Cu}$ samples.

Question/Comments 6: The theoretical calculations part should be extended. What happens to the energy gap with increasing Cu-content in Pb-Se-S? Show some electronic density of state, compare theoretically calculated E_g -values with those deduced from the Arrhenius plots to your $\sigma(T)$ data. The trends from theory and experiment are expected to agree with each other.

Reply: Thanks for your suggestion. As the Cu doping amount is quite small, a large supercell is required for the theoretical calculation, which is challenging to implement. To ensure feasibility, two simplified models were employed in theoretical calculation: $\text{Pb}_{27}\text{S}_{16}\text{Se}_{11}$ and $\text{Pb}_{27}\text{S}_{16}\text{Se}_{11}\text{Cu}$. The projected density of states (PDOS) of $\text{PbS}_{0.6}\text{Se}_{0.4}$ and $\text{PbS}_{0.6}\text{Se}_{0.4}+\text{Cu}$ were calculated respectively, as shown in Fig. R4(a) and (b). The calculated band gap of $\text{Pb}_{27}\text{S}_{16}\text{Se}_{11}$ and $\text{Pb}_{27}\text{S}_{16}\text{Se}_{11}\text{Cu}$ are very close, which is about 0.58 eV. These results show that the interstitial Cu have little impact on the band gap of Pb-S-Se system. Moreover, the adding Cu would make the Fermi level move toward the conduction band. The optical band gap of samples with different Cu content was also measured, as shown in Fig R4(c). Since the amount of Cu doping is very small, the optical band gap of all samples is almost constant, which is about 0.31eV. Additionally, we also plotted the $\ln p - T^{-1}$ of Cu doped samples. Fig. R4(d) shows that the material exhibits heavily doped semiconductor transport characteristics and have similar slopes, which means the energy band gap are basically unchanged. In conclusion, the trends observed in theory and experiment are found to be consistent.

The corresponding data and discussion had been added into the revised manuscript.

Figure R4 The projected density of states (DOS) of (a) $\text{Pb}_{27}\text{S}_{16}\text{Se}_{11}$ and (b) $\text{Pb}_{27}\text{S}_{16}\text{Se}_{11}\text{Cu}$. (c) Optical band gap test results and (d) $\ln I_0 - T^{-1}$ of $\text{Pb}_{1.004}\text{S}_{0.6}\text{Se}_{0.4} + z\text{Cu}$ ($z = 0 - 0.006$) samples.

Question/Comments 7: Fifty citations are obviously insufficient to reflect the state-of-the-art in the TE properties of lead chalcogenides. Authors have to analyze more carefully the literature data.

Reply: Thanks for your reminding. We have added some references.

Reviewer: 2

General Comment: In this work, lattice planification and interstitial doping strategies were employed to optimize the thermoelectric performance of n-type PbS crystal, resulting in a remarkable power generation efficiency of ~8% in the material. Additionally, this work demonstrates the first thermoelectric cooling device based on the n-type PbS material, which exhibits a decent cooling temperature difference of ~37 K at room temperature. This study shows the low-cost and earth-abundant PbS crystal could be a viable alternative to commercial n-type Bi₂Te₃ thermoelectric materials. I believe this work is sound and comprises substantial novelty and significance. Therefore, it is important to publish the present results as they provide new insights to the PbS thermoelectric material. I recommend this interesting work for publication after a minor revision.

Reply: Thanks for your positive comments, we appreciated your comments and we have provided a point-to-point response as follows.

Question/Comments 1: I did not find the fabrication process of the TE device in the experimental section. Please provide the details.

Reply: Thanks for your reminding. For the device fabrication, the optimized PbS ingot and commercial p-type Bi₂Te₃ were cut into cuboids with size of 2×2×4 mm³ and then put into ethyl alcohol for ultrasonic cleaning. Then the cuboids were galvanized to form a Ni-based barrier layer using a commercial nickel-plating solution, and then the cuboids were soldered onto Cu electrode using Sn₄₈Bi₅₂ solder paste. Finally, the n-type PbS-based thermoelectric cooling module was fabricated and the cooling performance was tested using Z-meter DX4090. We have completed the fabrication process in the experimental section.

Question/Comments 2: When comparing the TE performance of their PbS sample with other reported materials, the authors calculated the average ZT from 300 to 823 K, but for the average power factor, they only considered values from 300 to 523 K. Is there a specific reason for this difference? It should be explained.

Reply: Thanks for your comments. The high performance of our sample is attributed to the high PF in the near room temperature range. So, we show the calculated PF_{ave} from 300 to 523 K to more intuitively highlight our advantages at near room temperature. Besides, the comparison of power factor among our work and reported materials in the temperature range of 300 - 823 K is shown in Fig 5b.

Question/Comments 3: Why the Cu clusters look darker in the STEM image?

Reply: The Cu clusters appear darker in the STEM image due to the diffraction contrast. As Cu occupies the interstitial position, it causes little lattice distortion and diffract part of the electron beam. Consequently, the areas containing Cu atoms will appear darker than the region without interstitial atoms.

Question/Comments 4: There are few typos in the figure. In Fig.4h, the crystal sample has been labeled as “ingot 2”, it should be changed to “crystal”; In the legend of Fig 2f, the Pb content was label as“x”, while in other panels, it was named as“y”, please correct.

Reply: Thanks for your careful reading. We have corrected the legend shown in Fig. 2f (Fig. 2i) and Fig. 4h.

Reviewer: 3

In this work, the authors prepared a series of $\text{Pb}_{1.004}\text{S}_{0.6}\text{Se}_{0.4}+\text{Cu}$ samples and significantly improved their thermoelectric performance by employing the lattice planification and interstitial doping strategies. They found that, minor Pb compensation improves the crystal quality and enhances carrier mobility, while interstitial effectively increases the carrier density. These effects result in a high conversion efficiency of $\sim 8\%$ in the material. Furthermore, this work demonstrates the first PbS-based thermoelectric cooling device, which exhibits an impressive cooling temperature difference of ~ 36.9 at room temperature. All the data are analyzed properly, and the paper is well written. Thereby, I recommend this work for publication with minor revision. Additionally, I have a few suggestions which author should consider during revision:

Reply: Thanks for your constructive comments, we appreciated your comments and we have provided a point-to-point response as follows.

Question/Comments 1: The 0.6% Cu-doped sample exhibits the highest TE performance. What would happen if further increases in Cu content? Besides, the authors should clarify why they selected the 0.1% Cu-doped sample for device fabrication.

Reply: Thanks for your comments. As shown in Fig. 3i, the 0.6% doped sample exhibits the highest ZT value, but its room temperature ZT falls short due to the high carrier density. Further doping with Cu is projected to lead to a decrease in room temperature ZT and an increase in ZT at higher temperatures. Indeed, we have conducted the thermoelectric properties measurement of Cu doped samples with Cu content up to 1%, the ZT_{max} remained consistent with that of the 0.6% Cu-doped sample, but the room temperature ZT reduces to 0.2, as shown in Fig. R5. Given that the 0.1% Cu-doped sample shows highest ZT_{ave} across the observed temperature range and the highest power factor value at room temperature among the Cu doped samples, it was chosen to perform crystal growth and device fabrication.

Fig. R5. Temperature-dependent (a) electric conductivity, (b) Seebeck coefficient, (c) power

factor, (d) total thermal conductivity, (e) lattice thermal conductivity, (f) ZT of $\text{Pb}_{1.004}\text{S}_{0.6}\text{Se}_{0.4} + z\text{Cu}$ ($z = 0.006 - 0.010$) samples.

Question/Comments 2: More details about the fabrication process of the TE device, including steps such as material preparation, device assembly, and soldering procedure, should be provided in the methods section.

Reply: Thanks for your reminding. For the device fabrication, the optimized PbS ingot and commercial p-type Bi_2Te_3 were cut into cuboids with size of $2 \times 2 \times 4 \text{ mm}^3$ and then put into ethyl alcohol for ultrasonic cleaning. Then the cuboids were galvanized to form Ni-based barrier layer using commercial nickel-plating solution, and then the cuboids were welded on Cu electrode using $\text{Sn}_{48}\text{Bi}_{52}$ solder paste. Finally, the n-type PbS-based thermoelectric cooling module was fabricated and the cooling performance was test using Z-meter DX4090. The fabrication process has been completed in the experimental section.

Question/Comments 3: Why the lattice parameter did not change when adding extra Cu into the lattice. The authors should provide an explanation for this observation.

Reply: Thanks for your comments. Since Cu atoms occupy interstitial site and the Cu content is quite small, the Cu atom would not cause significant change in the lattice structure. So, the lattice parameter did not show significant change with the addition of small amounts of Cu.

Question/Comments 4: More details about the crystal preparation process should be provided, particularly regarding what happens after cooling the crystal sample at a rate of 1 K per h from 1403 K to 1303 K.

Reply: Thanks for your reminding. We have provided a more detailed description about the crystal preparation process in the experimental section. The crystal samples were obtained through the vertical Bridgman method, they were first synthesized with the same procedure as for polycrystal, then the obtained ingot was grounded into powder. The powder was loaded into silica tubes and flame sealed at a residual pressure under 10^{-4} Pa. Then it was heated to 1403 K in 11 h, kept at this temperature for 3 h and cooled at a rate of 1 K per h from 1403 K to 1303 K. Finally, the furnace was shut down and cooled to room temperature.

Question/Comments 5: The authors should correct any typos found in the paper.

Reply: Thanks for your reminding. We have carefully corrected the typos in the paper.

REVIEWERS' COMMENTS

Reviewer #1 (Remarks to the Author):

Authors improved the manuscript considerably and now it can be published in e.g. Chemistry of materials or a similar journal. As I wrote in the first review the novelty of the results provided in the current manuscript is insufficient to become published in Nature communication.

Reviewer #2 (Remarks to the Author):

All concerns have been resolved and no further revisions are needed. I agree to publish this paper.

Reviewer #3 (Remarks to the Author):

Attached.

Comment:

The revised manuscript has significantly improved the quality of the manuscript.
Thereby, I recommend this work for publication.